# Wastewater surveillance of SARS-CoV-2 genomic populations on a country-wide scale through targeted sequencing

Florencia Cancela[1], Natalia Ramos[1], Davida S. Smyth[2], Claudia Etchebehere[3], Mabel Berois[1], Jesica Rodríguez[4], Caterina Rufo[4], Alicia Alemán[5], Liliana Borzacconi[6], Julieta López[7], Elizabeth González[7], Germán Botto[8], Starla G. Thornhill[2], Santiago Mirazo[1,9]*, Mónica Trujillo[10]*

1 Sección Virología, Instituto de Química Biológica, Facultad de Ciencias, Universidad de la República, Montevideo, Uruguay, 2 Department of Life Sciences, Texas A&M University-San Antonio, San Antonio, Texas, United States of America, 3 Departamento de Bioquímica y Genómica Microbiana, Instituto de Investigaciones Biológicas Clemente Estable, Ministerio de Educación y Cultura, Montevideo, Uruguay, 4 Laboratorio de Alimentos y Nutrición, Polo Tecnológico de Pando, Facultad de Química, Universidad de la República, Montevideo, Uruguay, 5 Departamento de Medicina Preventiva, Facultad de Medicina, Universidad de la República, Montevideo, Uruguay, 6 Instituto de Ingeniería Química, Facultad de Ingeniería, Universidad de la República, Montevideo, Uruguay, 7 Departamento de Ingeniería Ambiental, Facultad de Ingeniería, Universidad de la República, Montevideo, Uruguay, 8 Departamento de Métodos Cuantitativos, Facultad de Medicina, Universidad de la República, Montevideo, Uruguay, 9 Departamento de Bacteriología y Virología, Instituto de Higiene, Facultad de Medicina, Universidad de la República, Montevideo, Uruguay, 10 Department of Biological Sciences and Geology, Queensborough Community College of The City University of New York, Queens, New York, United States of America

* mtrujillo@qcc.cuny.edu (MT); smirazo@higiene.edu.uy (SM)

**Data Availability Statement:** The raw sequence data has been deposited in the NCBI-SRA database under the BioSample accession numbers

## Abstract

SARS-CoV-2 surveillance of viral populations in wastewater samples is recognized as a useful tool for monitoring epidemic waves and boosting health preparedness. Next generation sequencing of viral RNA isolated from wastewater is a convenient and cost-effective strategy to understand the molecular epidemiology of SARS-CoV-2 and provide insights on the population dynamics of viral variants at the community level. However, in low- and middle-income countries, isolated groups have performed wastewater monitoring and data has not been extensively shared in the scientific community. Here we report the results of monitoring the co-circulation and abundance of variants of concern (VOCs) of SARS-CoV-2 in Uruguay, a small country in Latin America, between November 2020—July 2021 using wastewater surveillance. RNA isolated from wastewater was characterized by targeted sequencing of the Receptor Binding Domain region within the spike gene. Two computational approaches were used to track the viral variants. The results of the wastewater analysis showed the transition in the overall predominance of viral variants in wastewater from No-VOCs to successive VOCs, in agreement with clinical surveillance from sequencing of nasal swabs. The mutations K417T, E484K and N501Y, that characterize the Gamma VOC, were detected as early as December 2020, several weeks before the first clinical case was reported. Interestingly, a non-synonymous mutation described in the Delta VOC, L452R, was detected at a very low frequency since April 2021 when using a recently described sequence analysis tool (SAM Refiner). Wastewater NGS-based surveillance of

SAMN31036808-SAMN31036817 and BioProject number PRJNA884675.

**Funding:** Funding was provided by the National Center for Science and Civic Engagement to DSS and MT to support wastewater research in the global south. The grant supported the costs associated with the RNA extraction and sequencing. The funders had no role in study design, data collection and analysis, decision to publish, or preparation of the manuscript.

**Competing interests:** The authors have declared that no competing interests exist.

SARS-CoV-2 is a reliable and complementary tool for monitoring the introduction and prevalence of VOCs at a community level allowing early public health decisions. This approach allows the tracking of symptomatic and asymptomatic individuals, who are generally under-reported in countries with limited clinical testing capacity. Our results suggests that wastewater-based epidemiology can contribute to improving public health responses in low- and middle-income countries.

## Introduction

Early in the COVID-19 pandemic it was found that SARS-CoV-2 was shed in human stool, indicating that human wastewater may be one route by which the virus could be detected. Shortly thereafter, the genetic signal of SARS-CoV-2 was identified in wastewater in many European countries, and in the USA [1, 2]. Since then, SARS-CoV-2 wastewater surveillance (WWS), the measurement of pathogen burden in wastewater by RT-qPCR, has undergone considerable improvement mainly due to an optimization of the concentrating and detecting protocols [3–5]. WWS has been used to evaluate infection trends in the community, monitor public health interventions and to make timely, evidence-based decisions to mitigate the impact of epidemic waves or outbreaks [6]. Additionally, WWS can provide cost-effective support to conventional surveillance methodologies that heavily depend on testing, clinical examination, and case reporting systems.

Few countries in Latin America and the Caribbean regions have developed a WWS program despite the fact that these countries had some of the highest mortality rates. COVID-19 threatens to undo recent gains in human capital outcomes and are the ones that could benefit the most given the low-resource requirements in comparison with other surveillance approaches. WWS can not only provide information on the circulation of pathogens but also identify clusters of cases before levels of sustained transmission are reached, allowing public health professionals to implement targeted control measures [7]. Two years after the first detection of SARS-CoV-2 in wastewater, the World Health Organization (WHO) published guidelines for the WWS of SARS-CoV-2 as a complementary tool to molecular and clinical diagnostics, emphasizing the need of this method to improve surveillance and epidemiological modeling [8]. Since it was first characterized in 2019, SARS-CoV-2 has accumulated mutations that led to the emergence of several viral variants of interest (VOIs) and variants of concern (VOCs). These VOCs accumulated changes in the Receptor-Binding Domain (RBD) of the viral spike (S) protein (D614G, N501Y, E484K, L452R, K417N, and T478K amino acid replacements) that have been associated with an increased transmissibility, a reduction in neutralization by natural or vaccine-derived antibodies and to differences in the clinical outcomes [9–14]. The COVID-19 pandemic has surged in successive epidemic waves, typically associated with the emergence of different VOCs. No short or mid-term public health strategy has been successful in predicting and preventing infection waves [15, 16].

WWS has been used for identifying increasing trends of community-level infections and predicting epidemic peaks with a reasonable anticipation time, thus providing opportune public health interventions [17–21]. Furthermore, by the introduction of genome sequencing methodologies, WWS has expanded from detecting traces of viral RNA by RT-qPCR to tracking the introduction and prevalence of mutations associated with VOCs circulating in the community [22].

Uruguay is a small country in South America with 3 million inhabitants, bordering Brazil (terrestrial borders) and Argentina (maritime borders). The country is organized in 19

departments, with most of the population concentrated in Montevideo, where the capital of the country is located. The intense human mobility across the terrestrial border with Brazil presented a major challenge for public health authorities and clinical data has shown repeated entry of the virus into Uruguay and the subsequent emergence of local outbreaks in Uruguayan border localities [23]. In Uruguay, two major COVID-19 epidemics have been described since the first report of the disease in March 2020. The first wave occurred in April-June 2021 (Fig 1A) and was associated with the Gamma VOC (Fig 1B), while the most recent surge (December-January 2022) was linked to the Omicron VOC. Furthermore, the Delta and other VOIs were also detected [24]. Once the epidemic pattern of SARS-CoV-2 infections was established, characterized by successive waves of distinct VOCs and lineages, a concerted effort was made in Uruguay to establish a uniform program of genomic sequencing of genetic material from nasal swabs from clinical cases, that would allow for the investigation of viral variant distribution and changes over time. Though critically important to establish the epidemiological patterns, a focus on clinical sampling is limited for several reasons, most notably for its cost, which is prohibitive in resource limited communities, and in addition, clinical samples are skewed towards the most symptomatic individuals generating a biased sample [25].

Uruguay established a consistent program of clinical testing that mostly focused on symptomatic patients and their contacts (test-trace-isolate strategy) [26]. Additionally, a program of genomic surveillance investigated the SARS-CoV-2 variant dynamics and distribution over time [24]. The goal of SARS-CoV-2 surveillance using wastewater goes beyond the detection of the genetic signal of the virus it includes monitoring the different viral lineages and their distribution over time. This has been successfully done across the world [27–29] using whole genome sequencing and/or targeted sequencing. Wastewater contains virions shed by numerous infected individuals; therefore, individual mutations cannot be reliably assigned to a specific genome. To overcome these limitations a targeted sequencing approach [28, 30–32] was chosen. Using a similar targeted approach to amplify and sequence a portion of the RBD of SARS-CoV-2 as a single amplicon, we were able to characterize the viral lineages present in sewersheds. Additionally, this strategy allowed us to measure the abundances of the different VOCs present in the community and to identify the introduction of novel variants and VOCs prior to their detection in clinical case-based genomic surveillance.

## Materials and methods

A graphical overview of the processing and analysis of the wastewater samples is presented in S1 Fig.

### Wastewater sampling and treatment plants

Two different and complementary sampling strategies were carried out. In one, sampling was done once a month from December 2020 to July 2021 (8 samples in total) at a wastewater treatment plant in Rivera, a city with very intense human mobility across the border with Brazil. During the first week of July 2021 wastewater samples were collected in four cities of Uruguay.

The selected cities were: Montevideo, the capital city of Uruguay, with the largest population of the country (about 1.3 million inhabitants); Salto, a border city with Argentina (104011 inhabitants); Rivera, border city with Brazil (64465 inhabitants); and Castillos an eastern seaside city from Rocha department (7541 inhabitants). An additional sample, collected in June 2021 in the Melo city wastewater treatment plant (51830 inhabitants), was also included.

Montevideo city has a combined sewer system that collects rainwater runoff, domestic sewage, and industrial wastewater into one pipe. All the water is transported into a wastewater treatment plant for pre-treatment, then is discharged to an underwater outfall in the Rio de la Plata.

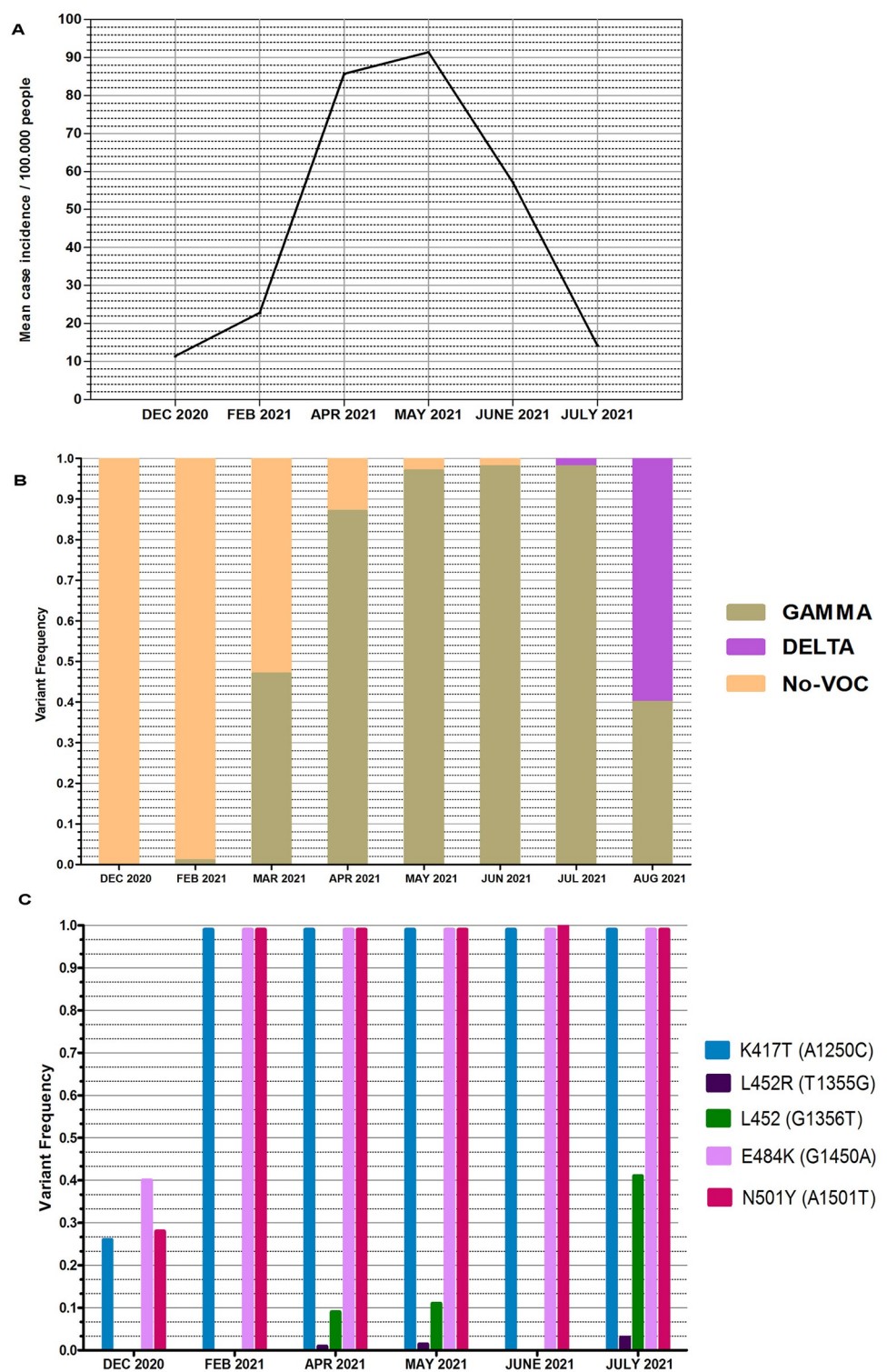

**Fig 1. Sampling timeline from Rivera city between December 2020 to July 2021. A**) Mean case incidence per 100,000 people from December 2020 to July 2021. **B**) Variant frequency of Gamma, Delta and No-VOC variants in Uruguay between December 2020 and November 2021 according to published epidemiological data. (Ministerio de Salud Pública, 2022). **C**) Variant frequency of the amino acid changes identified in each wastewater sample.

Salto, Castillos, Melo and Rivera have a different system in which domestic sewage and rainwater runoff do not mix. However, the system is leaky as rainwater does enter the sewage collector and the effluent is consequently diluted when precipitation occurs.

## Wastewater concentration

The wastewater concentration method was adapted from a previously published protocol [33]. Briefly, 45 mL of wastewater samples were centrifuged at 4,750 g for 30 minutes. Supernatant was then removed carefully without disturbing the pellet. 40 mL of supernatant was mixed with PEG 8000 (0.1 g/mL) and NaCl (0.0225 g/mL), shaken at room temperature for 15 minutes and centrifuged at 12,000 g (Sigma 2-16KL) for 2 hours at 4°C. After centrifugation, the supernatant was discarded, and samples were centrifuged again for 5 minutes at 12,000 g to remove the remaining supernatant. Viral pellets were resuspended in 140 μL RNase-free Phosphate Buffered Saline (PBS).

## Viral extraction, detection, and quantification

Viral RNA was extracted from wastewater concentrates using the QIAamp® Viral RNA Mini Kit (Qiagen) following manufacturer's instructions. RNA concentrations were determined using a NanoDrop™ Lite Spectrophotometer (Thermo Scientific, USA). SARS-CoV-2 RNAs were detected using the kit COVID-19 RT-qPCR Real TM Ambiental (*Cy5)* (ATGen S.R.L., Montevideo, Uruguay). The kit is a one-step, reverse transcription-Real Time PCR, that targets the N (nucleocapsid) gene of the SARS-CoV-2. Additionally, the kit contains primers and probes for the amplification of an internal control that is added during the RNA extraction step, which constitutes an RNA extraction control. Amplification was performed in a 20 μL reaction mixture containing 10 μL of extracted RNA from each sample. A positive control was provided with the kit, and nuclease free water was used as non-template control. RT-qPCR steps consisted of reverse transcription at 50°C for 5 min and activation at 95°C for 20 s, followed by 40 cycles of denaturation at 95°C for 15 s followed by annealing/extension at 60°C for 30s. The reactions were performed using a Rotor GeneQ (Qiagen, USA). Real-time PCR Ct value, date and sampling place from Uruguayan wastewater samples are reported in S1 Table.

## RBD PCR amplification

The RNAs from the positive samples were reverse transcribed using SuperScript™ IV Reverse Transcriptase and random primers (LifeTechnologies, USA). The RBD region was amplified using Q5® High-Fidelity DNA Polymerase (New England Biolabs, M0491S) [28]. PCR was performed as follows: 98°C for 1 min, 40 cycles of 98°C for 15 s, 58°C for 15 s and 68°C for 1 min; finally, an extension at 68°C for 2 min. Amplicons of the expected size (330 bp) were gel-purified with PureLink™ Quick Gel Extraction Kit (Life Technologies, USA).

## Ion Torrent next generation sequencing (NGS)

Sequencing libraries were produced from 150 ng purified PCR products using NEBNext® Fast DNA Library Prep Set for Ion Torrent™ (#E6270L, New England Biolabs, Inc) and Ion Xpress™ Barcode Adapters (#4474517, ThermoFisher Scientific). Libraries were quantified using Qubit™ Fluorometer (Qubit™ dsDNA HS Assay Kit), and 50 pM inputs were used for the sequencing template preparation (400bp, Ion Chef protocol) and chip loading in a Ion Chef™ Instrument (#A34019, Ion 510™ & Ion 520™ & Ion 530™ Kit–Chef, ThermoFisher Scientific). Sequencing was performed in the Ion GeneStudio™ S5 System™ at the NGS sequencing

platform of Instituto de Investigaciones Biológicas Clemente Estable (Montevideo, Uruguay). All procedures were according to manufacturer instructions.

### NGS data analysis

Analysis of the data obtained included the adapter/quality trimming that was performed with Trim Galore considering base quality >25.

The reference alignment was carried out with the BWA-MEM program and the trimmed reads mapped to two reference sequences of SARS-CoV-2 (Genbank accession number OK091006 and MN908947, corresponding to Delta VOC and Wuhan strain respectively). Consensus calling was obtained with Samtools and IVAR and the spectra of variants in wastewater samples was analyzed with the LoFreq tool [34]. Furthermore, trimmed reads were dereplicated to unique sequences and then mapped to Delta and Wuhan reference sequences of SARS-CoV-2 using BWA-MEM.

To compare the results obtained with an additional approach, mapped RBD amplicons were processed with SAM Refiner [31] to call variants and their abundance.

To identify frequency variants, present in the filtered mapped reads, two requirements were considered: i) sequencing coverage > 100 and ii) variant frequency ≥ 5%.

Graphical analysis was performed with GraphPad Prism v5.

### iSeq sequencing (NGS)

An aliquot of the cDNA synthesized from RNA isolated from wastewater was further processed for iSeq sequencing of a region within the viral RBD in San Antonio, Texas. This RBD region was amplified using Q5$^®$ High-Fidelity DNA Polymerase (New England Biolabs, M0491S) using primers that incorporate Illumina adapters [30]. PCR performed as follows: 98˚C for 30 s plus 40 cycles of 98˚C for 5 s, 53˚C for 15 s and 65˚C for 1 min; finally, an extension of 65˚C for 1 min.

The RBD amplicons were purified using AMPure XP beads (Beckman Coulter, A63881). Index PCR was performed using the Nextera DNA CD Indexes kit (Illumina, 20018707) with 2X KAPA HiFi HotStart ReadyMix (Roche, KK2601), and indexed PCR products purified using AMPure beads (Beckman Coulter, A63881). The indexed libraries were quantified using the Qubit 3.0 and Qubit dsDNA HS Assay Kit (Invitrogen, Q32854) and diluted in 10 mM Tris-HCl to a final concentration of approximately 0.3 ng/μL (1 nM). The libraries were pooled together and diluted to a final concentration of 50 pM. Before sequencing on an Illumina iSeq100, a 10% spike-in of 50 pM PhiX control v3 (Illumina, FC-110-3001) was added to the pooled library. The Illumina iSeq instrument was used to generate paired-end 150 base pair length reads. iSeq reads were uploaded to the BaseSpace Sequence Hub and demultiplexed using a FASTQ generation script. Reads were processed using the published Geneious workflows [35] for preprocessing of NGS reads and assembly of SARS-CoV-2 amplicons variants were called using the Annotate and Predict Find Variations/SNPs in Geneious and verified by using the V-PIPE SARS-CoV-2 application [36]. Paired reads were trimmed, and the adapter sequences removed with the BBDuk plugin. Trimmed reads were aligned to the SARS-CoV-2 reference genome MN908947.

### Clinical cases

All clinical data was obtained from public dashboards from the Ministerio de Salud Publica and Instituto Pasteur de Montevideo. All dashboard data is in aggregated format.

## Results

### Viral RNA detection and RBD Ion Torrent NGS sequencing

After viral concentration, SARS-CoV-2 RNA was detected by RT-qPCR in 12 wastewater samples, with Ct values ranging from 27 to 32.47 (S1 Table). The RBD region amplified in this study included the aa positions 417, 452, 484 and 501 known to be mutated among several VOCs. Substitutions at these positions enabled us to identify SARS-CoV-2 circulating variants in wastewater. A total of 10,849–110,597 sequencing raw reads were obtained for 10 out of 12 samples analyzed. Two samples from Rivera city (from January and March) could not be sequenced. Table 1 summarizes the nucleotide, amino acid changes and their frequencies in the PCR amplicons (compared to Delta reference sequence) using the LoFreq tool. Similar results were obtained by using Wuhan reference sequence NC_045512 for comparison.

The raw sequence data has been deposited in the NCBI-SRA database under the BioSample accession numbers SAMN31036808-SAMN31036817 and BioProject number PRJNA884675.

### Analysis of samples from Rivera city

Starting from December 2020, the VOC-associated mutations K417T, E484K and N501Y were detected albeit at a low frequency. However, from February to July 2021 the aa substitutions K417T, E484K and N501Y became the dominant ones with a very high frequency, indicating the presence of the Gamma VOC. Furthermore, in the samples from April and May, a synonymous substitution was identified at position G1356T (L452) with frequencies of 8% and 11%, respectively (Fig 1C). Interestingly, according to the monthly epidemiological reports from the Ministerio de Salud Pública, the Gamma VOC was not detected in clinical specimens until March (Fig 1B).

### Analysis of samples for June- July 2021

To take a snapshot of the VOCs distribution across the country for July 2021 samples from four cities (Montevideo, Rivera, Salto and Castillos) were sequenced (Fig 2). An additional sample corresponding to June 2021 from Melo was included.

In all the cases, the three mutations K417T, E484K and N501Y were present at a similar high frequency, above 97%.

Samples from Rivera exhibited greater heterogeneity and had more mutations. Particularly, the synonymous substitution G1356T (L452), which had been detected several months ago, reached a frequency of 41%.

The results of using SAM Refiner to monitor SARS-CoV-2 lineages in wastewater are shown in Table 2. This tool enabled us to further characterize very low frequency variants like the L452R mutation (corresponding to the Delta variant). Samples from Rivera, April, May, and July 2021 contained the L452R amino acid substitution with 1%, 1.4% and 3% abundance respectively.

### RBD iSeq NGS sequencing

To further analyze the samples from Uruguay, cDNA was sent to San Antonio, Texas for processing on the iSeq100 pipeline as had been done previously [28]. Challenges in shipping resulted in the package arriving broken and at room temperature. As a result, a wide variety in read quality and quantity was observed for the samples with as few as 54 aligned reads to as many as 47,872 being obtained. Only samples with more than 100 reads were analyzed. It was possible to identify several notable SNPs in the aligning reads including L452R (samples 0 and 7), E484K (samples 0, 7 and 12), N501Y (samples 0, 7 and 12), S494P (sample 0). This

**Table 1. Analysis using the LoFreq tool.**

| Sample Number | Date | Place | Predominant Variant | Nt change (Spike) | AA Change | Variant frequency | Mapped reads | Coverage |
|---|---|---|---|---|---|---|---|---|
| 13 | Dec 2020 | Rivera | No VOC* | A1250C | K417T | 0.258451 | 7143 | 10029 |
| | | | | G1450A | E484K | 0.400333 | | 6015 |
| | | | | A1501T | N501Y | 0.277101 | | 6402 |
| 14 | Feb 2021 | Rivera | Gamma | A1250C | K417T | 0.991176 | 15137 | 13712 |
| | | | | G1450A | E484K | 0.999021 | | 8175 |
| | | | | A1501T | N501Y | 0.999050 | | 8424 |
| 15 | Apr 2021 | Rivera | Gamma | A1250C | K417T | 0.986661 | 5510 | 5023 |
| | | | | G1356T | None | 0.089392 | | 2696 |
| | | | | G1450A | E484K | 0.998507 | | 2679 |
| | | | | A1501T | N501Y | 0.998625 | | 2910 |
| 16 | May 2021 | Rivera | Gamma | A1250C | K417T | 0.989572 | 4957 | 4507 |
| | | | | G1356T | None | 0.114674 | | 2869 |
| | | | | G1450A | E484K | 0.998602 | | 2861 |
| | | | | A1501T | N501Y | 0.998326 | | 2987 |
| 17 | June 2021 | Rivera | Gamma | A1250C | K417T | 0.990765 | 4888 | 4548 |
| | | | | G1450A | E484K | 0.999242 | | 2640 |
| | | | | A1501T | N501Y | 1.000000 | | 2665 |
| 45 | June 2021 | Melo | Gamma | A1250C | K417T | 0.972526 | 52906 | 49246 |
| | | | | G1450A | E484K | 0.997796 | | 48100 |
| | | | | A1501T | N501Y | 0.997755 | | 47654 |
| 18 | July 2021 | Rivera | Gamma | A1250C | K417T | 0.994018 | 6232 | 5684 |
| | | | | G1356T | None | 0.408698 | | 4047 |
| | | | | G1450A | E484K | 0.999259 | | 4046 |
| | | | | A1501T | N501Y | 0.999282 | | 4176 |
| 0 | July 2021 | Montevideo | Gamma | A1250C | K417T | 0.991028 | 5635 | 5127 |
| | | | | G1450A | E484K | 0.998639 | | 2938 |
| | | | | A1501T | N501Y | 0.999684 | | 3160 |
| 7 | July 2021 | Castillos | Gamma | A1250C | K417T | 0.977448 | 3708 | 3370 |
| | | | | G1356T | None | 0.012235 | | 2121 |
| | | | | G1450A | E484K | 0.978017 | | 2040 |
| | | | | A1501T | N501Y | 0.981375 | | 2094 |
| 10 | July 2021 | Salto | Gamma | A1250C | K417T | 0.987458 | 21690 | 19455 |
| | | | | G1450A | E484K | 0.991361 | | 12038 |
| | | | | A1501T | N501Y | 0.992046 | | 12447 |

The nucleotide and amino acid changes by Ion Torrent sequencing, variant and coverage calculated using the LoFreq tool. The nucleotide and amino acid changes from wastewater amplicons in Uruguayan cities by Ion Torrent sequencing compared to delta reference sequence (Genbank accession number OK091006.1) are shown. No VOC*: AA changes associated with VOC are present at frequencies >5%, however the VOC is not the predominant variant.

approach detected the same amino acid changes that the Ion Torrent NGS sequencing as shown in Table 3.

## Discussion

Wastewater detection of SARS-CoV-2 as a component of wastewater-based epidemiology (WWBE) has been demonstrated to be a useful, cost-effective, and reliable tool for monitoring COVID-19 dynamics, the emergence of local viral variants and the introduction of VOCs from other countries [37, 38]. Border crossings presented unique challenges during the

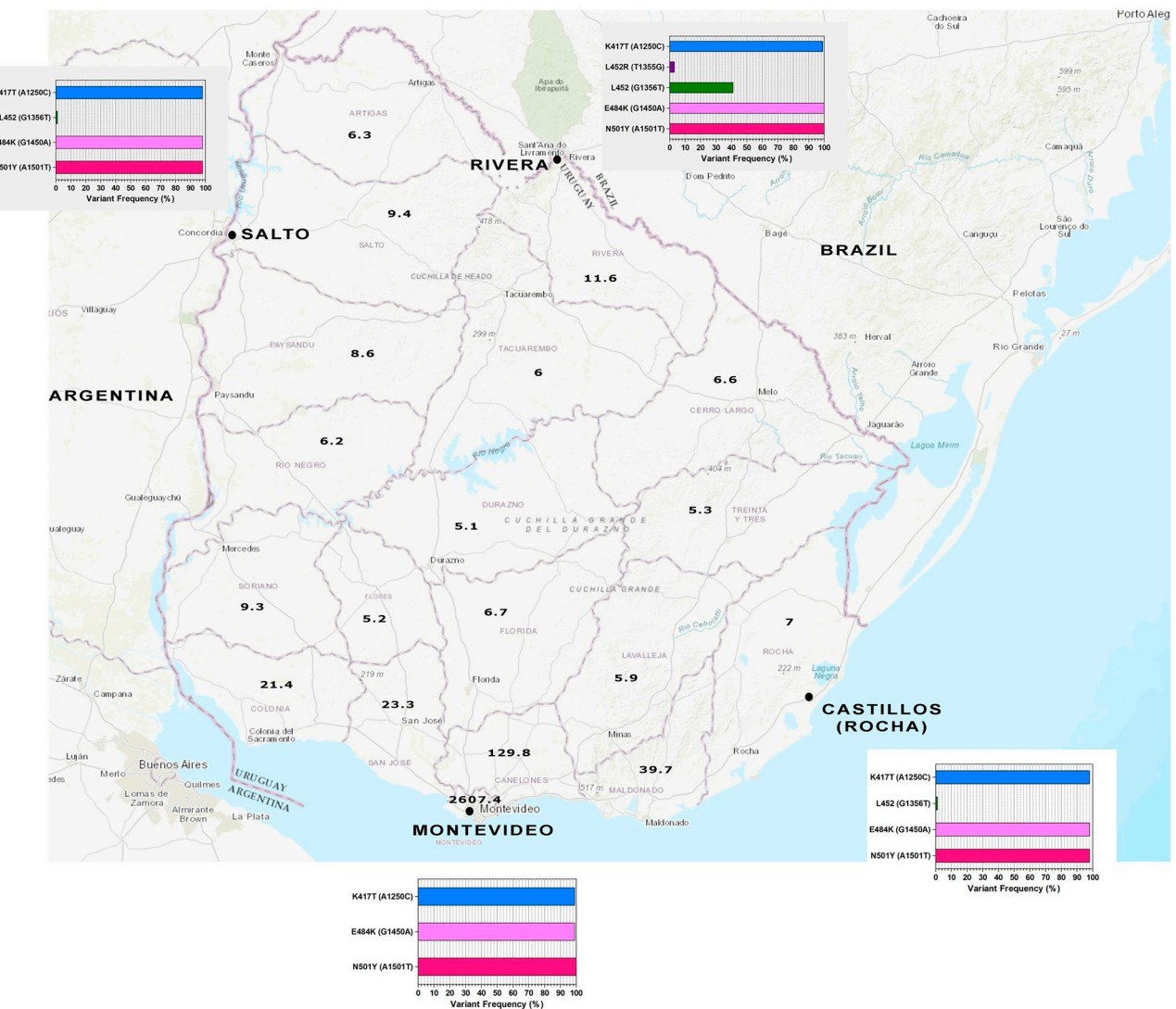

**Fig 2. Wastewater sampling from July 2021 in Montevideo, Rivera, Salto, and Castillos.** The amino acid changes identified, and their frequencies are shown in graphs for each location. Population density (People/Sq Km) per department are indicated in the map, based on the values extracted from Anuario Estadistico Nacional 2019, 96ª version, Instituto Nacional de Estadística (INE), www.ine.gub.uy. The map of Uruguay was obtained from USGS National Map Viewer (http://viewer.nationalmap.gov/viewer/).

pandemic and informal border crossings such as those occurring across dry frontiers are hard to capture using clinical testing [39]. Uruguay has a ~1,100 km uninterrupted dry frontier with Brazil and although Uruguayan borders were closed, there was a flow of people across the border [23].

In this study, we monitored the SARS-CoV-2 genetic signal from wastewater in several regions of Uruguay. A targeted-sequencing approach, focused on the RBD of the S protein, allowed us to identify circulating VOCs in different stages of the epidemic outbreak.

In Uruguay, the first COVID-19 case was detected on March 13rd, 2020, and restrictive measures including restriction of social gathering, closing of school and borders were immediately established. Thus, though several local outbreaks occurred, epidemic waves were avoided until October 2021, when cases started to increase steadily [40, 41]. Most of the earlier outbreaks were reported from Rivera, a city bordering Brazil. In June 2021, Brazil was the country

**Table 2. Analysis done using the SAMRefiner tool.**

| Sample Number | Date | Place | Predominant Variant | Nt change (Spike) | AA Change | Variant frequency | Mapped reads | Coverage |
|---|---|---|---|---|---|---|---|---|
| 13 | Dec 2020 | Rivera | No VOC* | A1250C | K417T | 0.258450 | 3872 | 10029 |
| | | | | G1450A | E484K | 0.351339 | | 6015 |
| | | | | A1501T | N501Y | 0.259874 | | 6402 |
| 14 | Feb 2021 | Rivera | Gamma | A1250C | K417T | 0.991175 | 3421 | 13712 |
| | | | | G1450A | E484K | 0.999021 | | 8175 |
| | | | | A1501T | N501Y | 0.999050 | | 8424 |
| 15 | Apr 2021 | Rivera | Gamma | A1250C | K417T | 0.986661 | 1465 | 5023 |
| | | | | T1355G | L452R | 0.009637 | | 2698 |
| | | | | G1356T | None | 0.089392 | | 2696 |
| | | | | G1450A | E484K | 0.998507 | | 2679 |
| | | | | A1501T | N501Y | 0.998625 | | 2910 |
| 16 | May 2021 | Rivera | Gamma | A1250C | K417T | 0.989572 | 1560 | 4507 |
| | | | | T1355G | L452R | 0.013598 | | 2868 |
| | | | | G1356T | None | 0.114674 | | 2869 |
| | | | | G1450A | E484K | 0.998602 | | 2861 |
| | | | | A1501T | N501Y | 0.998326 | | 2987 |
| 17 | June 2021 | Rivera | Gamma | A1250C | K417T | 0.990765 | 1384 | 4548 |
| | | | | G1450A | E484K | 0.999242 | | 2640 |
| | | | | A1501T | N501Y | 1.000000 | | 2665 |
| 45 | June 2021 | Melo | Gamma | A1250C | K417T | 0.972526 | 14513 | 49246 |
| | | | | G1450A | E484K | 0.998025 | | 48100 |
| | | | | A1501T | N501Y | 0.997755 | | 47654 |
| 18 | July 2021 | Rivera | Gamma | A1250C | K417T | 0.994018 | 2059 | 5684 |
| | | | | T1355G | L452R | 0.032362 | | 4048 |
| | | | | G1356T | None | 0.408698 | | 4047 |
| | | | | G1450A | E484K | 0.999258 | | 4045 |
| | | | | A1501T | N501Y | 0.999282 | | 4176 |
| 0 | July 2021 | Montevideo | Gamma | A1250C | K417T | 0.991028 | 1607 | 5127 |
| | | | | G1450A | E484K | 0.998979 | | 2938 |
| | | | | A1501T | N501Y | 0.999683 | | 3160 |
| 7 | July 2021 | Castillos | Gamma | A1250C | K417T | 0.977448 | 1332 | 3370 |
| | | | | G1356T | None | 0.014131 | | 2123 |
| | | | | G1450A | E484K | 0.963890 | | 2047 |
| | | | | A1501T | N501Y | 0.981375 | | 2094 |
| 10 | July 2021 | Salto | Gamma | A1250C | K417T | 0.987458 | 5249 | 19455 |
| | | | | G1450A | E484K | 0.991360 | | 12038 |
| | | | | A1501T | N501Y | 0.992046 | | 12447 |

Nucleotide and amino acid changes by Ion Torrent sequencing, variant and coverage calculated using the SAMRefiner tool. The nucleotide and amino acid changes from wastewater amplicons in Uruguayan cities by Ion Torrent sequencing compared to delta reference sequence (Genbank accession number OK091006.1) are shown. No VOC*: AA changes associated with VOC are present at frequencies >5%, however the VOC is not the predominant variant.

with the second-highest rate of deaths per million and the fourth by the number of cases per million in Latin America and the Caribbean [42, 43].

Immediately, this city became the main concern for the government, and it was identified as a potential route for the introduction of active, imported COVID-19 cases and SARS-CoV-2 lineages.

**Table 3. Analysis using Geneious.**

| Sample Number | Date | Place | Predominant Variant | Nt change (Spike) | AA Change | Variant frequency | Mapped reads | Coverage |
|---|---|---|---|---|---|---|---|---|
| 0 | July 2021 | Montevideo | No VOC* | T1355G | L452R | 0.117 | 47872 | 24310 |
| | | | | G1450A | E484K | 0.062 | | 23064 |
| | | | | T1480C | S494P | 0.115 | | 23227 |
| | | | | A1501T | N501Y | 0.28 | | 23014 |
| 7 | July 2021 | Castillos | Gamma | T1355G | L452R | 0.031 | 2466 | 1210 |
| | | | | G1450A | E484K | 0.956 | | 1229 |
| | | | | A1501T | N501Y | 0.96 | | 1224 |
| 12 | July 2021 | Cerro Largo | No VOC* | G1450A | E484K | 0.174 | 38121 | 18429 |
| | | | | A1501T | N501Y | 0.435 | | 18382 |

Nucleotide and amino acid changes by Illumina iSeq sequencing, variant and coverage calculated using the Annotate and Predict Find Variations/SNPs in Geneious and verified by using the V-PIPE SARS-CoV-2 application (36). The nucleotide and amino acid changes from wastewater in Uruguayan cities by Illumina iSeq sequencing compared to Wuhan reference sequence (Genbank accession number MN908947) are shown.

No VOC*: AA changes associated with VOC are present at frequencies >5%, however the VOC is not the predominant variant.

In Rivera, qPCR data showed that Ct values in wastewater samples remained quite stable from December to July 2021, despite the cases/million people incidence rate exhibiting clear fluctuations in Uruguay during that period. This discrepancy can be explained by the lack of a reliable internal indicator to calculate the viral RNA recovery efficiency from wastewater. Limitations of this study also include the population mobility that characterizes border towns that are always difficult to measure even more during the implementation of pandemic restrictions.

The results from the targeted amplicon-NGS sequencing of the December samples from Rivera showed early evidence of the presence, though at low frequency, of several mutations associated to the Gamma VOC (K417T, E484K and N501Y), which were then confirmed in February when this variant became dominant in wastewater in that city. The Gamma VOC was identified in Manaos, Brazil, in November 2020 and then spread throughout the country within weeks [44]. Our findings agree with recent reports, based on phylogeographic analyses, that propose the outbreaks in Uruguay were mostly caused by introductions from neighboring South American countries, particularly Brazil [45, 46]. During this time an additional variant, named P.2 and classified as a VOI, was introduced to Uruguay, and detected in clinical swabs. Of note, this variant shared the E484K mutation with the Gamma variant [47], so they would not be differentiated using our approach.

Interestingly, the epidemiological data from the nasopharyngeal swabs samples in Uruguay revealed that the Gamma VOC was not detected in human cases until March. However, it was detected in wastewater in Rivera city, albeit at a low frequency; and by April it had become dominant (Fig 1C). Furthermore, a Bayesian analysis reported by Rego et al. [46] estimated the median time of the most recent common ancestor of the Gamma clades identified in COVID-19 cases in Uruguay to be mid-February to early March 2021.

Even though the Gamma VOC fueled the first epidemic wave in Uruguay, additional viral mutations were detected in human clinical samples that could not be detectable/identified by our targeted amplicon-approach. However, the exact role of these variants in the epidemic dynamics is unclear since they were rapidly displaced by Gamma [48].

In late 2020, the B.1.617 VOC (named Delta) was first detected in India and by July 2021 it had been reported in 60 countries [49]. To assess whether this VOC was already circulating in our country, wastewater was sampled in the first week of July from 4 regions. No evidence of

the Delta VOC associated mutations were detected. However, a more detailed analysis performed with SAM Refiner, a recently developed bioinformatic tool, showed that the L452R mutation was indeed present by April 2021 in Rivera (Table 2). Although we set a cut-off criterion of variant frequency $\geq 5\%$, we decided to show these data as it is considered a Delta VOC marker [50]. Interestingly, according to reports from the Ministerio de Salud Pública, the first clinical case associated with Delta infection dated from mid-July, and by August circulation at the community level had not been observed yet [51].

In this work, Spike targeted-sequencing enabled the early identification of SARS-CoV-2 VOCs circulation in wastewater samples, even at low frequencies and prior in advance of their detection in clinical samples.

Even though the number of mapped reads were modest, the results of VOCs variant frequencies and counts obtained by both bioinformatic approaches highly correlated, indicating a robust and reliable analysis.

Frequency analyses evidenced the rapid substitution of No-VOC lineages by the Gamma variant during the December-February 2020 period. The Gamma VOC likely originated in Rivera city and subsequently spread to the rest of the territory. Our analysis did not identify any variants/mutations in wastewater that had not been reported in clinical cases.

In summary, this study supports the use of wastewater surveillance as a reliable and complementary tool for monitoring the introduction, prevalence and spread of VOC and VOI at the community level. In particular, the early detection of new variants, before their detection in patients, can help local authorities to make informed public health decisions supporting better health outcomes.

## Supporting information

**S1 Fig. Graphical overview of the processing and analysis of the wastewater samples.**
(TIF)

**S1 Table. Real-time PCR Ct value, date, and sampling place from Uruguayan wastewater samples.**
(DOCX)

## Acknowledgments

We are very grateful to Intendencia Municipal de Montevideo (UODF—SOMS—Div. Saneamiento) and Obras Sanitarias del Estado for providing the samples. We also would like to thank Dr. Devon A. Gregory for his assistance in the use of the SAM Refiner program.

## Author Contributions

**Conceptualization:** Davida S. Smyth, Santiago Mirazo, Mónica Trujillo.

**Data curation:** Florencia Cancela, Natalia Ramos, Davida S. Smyth, Germán Botto, Starla G. Thornhill, Santiago Mirazo.

**Formal analysis:** Florencia Cancela, Natalia Ramos, Caterina Rufo, Starla G. Thornhill, Mónica Trujillo.

**Funding acquisition:** Davida S. Smyth, Mónica Trujillo.

**Investigation:** Florencia Cancela, Davida S. Smyth, Claudia Etchebehere, Mabel Berois, Jesica Rodríguez, Caterina Rufo, Alicia Alemán, Liliana Borzacconi, Julieta López, Elizabeth González, Germán Botto, Santiago Mirazo, Mónica Trujillo.

**Methodology:** Davida S. Smyth, Santiago Mirazo, Mónica Trujillo.

**Supervision:** Davida S. Smyth, Santiago Mirazo.

**Writing – original draft:** Davida S. Smyth, Santiago Mirazo, Mónica Trujillo.

**Writing – review & editing:** Florencia Cancela, Davida S. Smyth, Mabel Berois, Starla G. Thornhill, Santiago Mirazo, Mónica Trujillo.

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
