## [Decision Letter · Decision Letter 0]

21 Feb 2023

PONE-D-22-31923Wastewater surveillance of SARS-CoV-2 genomic populations on a country-wide scale through targeted sequencingPLOS ONE

Dear Dr. Trujillo,

Thank you for submitting your manuscript to PLOS ONE. After careful consideration, we feel that it has merit but does not fully meet PLOS ONE’s publication criteria as it currently stands. Therefore, we invite you to submit a revised version of the manuscript that addresses the points raised during the review process.

We look forward to receiving your revised manuscript.

Kind regards,

Nagarajan Raju

Academic Editor

PLOS ONE

Journal Requirements:

Funding was provided by the National Center for Science and Civic Engagement to DSS and MT to support wastewater research in the global south. The grant supported the costs associated with the RNA extraction and sequencing. 

3. We note that Figure 2 in your submission contain [map/satellite] images which may be copyrighted. All PLOS content is published under the Creative Commons Attribution License (CC BY 4.0), which means that the manuscript, images, and Supporting Information files will be freely available online, and any third party is permitted to access, download, copy, distribute, and use these materials in any way, even commercially, with proper attribution. For these reasons, we cannot publish previously copyrighted maps or satellite images created using proprietary data, such as Google software (Google Maps, Street View, and Earth). For more information, see our copyright guidelines: http://journals.plos.org/plosone/s/licenses-and-copyright.

Additional Editor Comments:

I suggest authors to go through the comments from the reviewers and resubmit the revised version of the manuscript.

Reviewers' comments:

Reviewer's Responses to Questions

**Comments to the Author**

1. Is the manuscript technically sound, and do the data support the conclusions?

Reviewer #1: Yes

Reviewer #2: Yes

2. Has the statistical analysis been performed appropriately and rigorously? 

Reviewer #1: N/A

Reviewer #2: N/A

3. Have the authors made all data underlying the findings in their manuscript fully available?

Reviewer #1: Yes

Reviewer #2: No

4. Is the manuscript presented in an intelligible fashion and written in standard English?

Reviewer #1: Yes

Reviewer #2: Yes

5. Review Comments to the Author

Reviewer #1: The country-wide SARS-CoV-2 surveillance of viral populations from wastewater samples is overall a promising and interesting study. The authors performed sampling and extraction and quantification of SARS-CoV-2 from different sampling sites across Uruguay at different time points. The manuscript is well-written and presented with appropriate figures and tables. However, there are some concerns which need to be addressed.

Major comment:

Wastewater surveillance was found to be a powerful and complementary approach to clinical testing in monitoring the SARS-CoV-2 pandemic. Authors report surveillance of variants of concern (VOCs) or No-VOCs agrees with clinical testing. It would be interesting to correlate these temporal changes with clinical data or the confirmed cases with statistical comparisons.

Figure 2 amino acids and their frequencies are shown in circular graphs for each location. Is this a Pie chart? Presented in the form of a Pie should sum to 100%.

Minor comment:

The scale on the Y-axis for Figure 1C is not completely visible.

Reviewer #2: The manuscripts presents useful results detailing wastewater surveillance (WWS) conducted using targeted amplification. It is well written.

It is somewhat surprising to see OK091006 used as a reference. While OK091006 indeed represents an example of the Delta variant of SARS-CoV-2 it is a Genbank record without extensive information on its provenance and quality control. The authors do not explain why this was chosen in addition to the more conventional Wuhan Hu-1 reference (which has been the subject of several revisions and has been deposited in NCBI's RefSeq coollection of curated sequences)

Finally the authors claim that data is available in NCBI SRA but I could not find it using the accession numbers provided. If the data is embargoed pending publication the authors should state that.

6. PLOS authors have the option to publish the peer review history of their article (what does this mean?). If published, this will include your full peer review and any attached files.

Reviewer #1: No

Reviewer #2: No

---

## [Author Response · Author response to Decision Letter 0]

26 Mar 2023

Thanks for your comments and suggestions. Please find below a detailed response to your points. This now includes the requests from the last email we received. 

1- We have introduced needed changes in formatting to meet PLOS ONE's style requirements.

The size of the titles for each section have been updated and the files have been labeled correctly. 

2. Please provide additional details regarding participant consent. In the Methods section, please ensure that you have specified (1) whether consent was informed and (2) what type you obtained (for instance, written or verbal). If your study included minors, state whether you obtained consent from parents or guardians. If the need for consent was waived by the ethics committee, please include this information.

None of our studies included human subjects. The only samples obtained by the authors were wastewater samples. All the clinical data obtained was in an aggregated format from public dashboards. We have included a sentence regarding this issue in the Methods section of the paper. 

Our ethics statement now indicates we did not handle any samples.

2- We have described the role of the funders:

Funding was provided by the National Center for Science and Civic Engagement to DSS and MT to support wastewater research in the global south. The grant supported the costs associated with the RNA extraction and sequencing. The funders had no role in study design, data collection and analysis, decision to publish, or preparation of the manuscript.

3- We have changed Figure 2. 

We have removed the original map. The Population Density per department indicated in the map is from the Anuario Estadistico Nacional 2019, 96a version, Instituto Nacional de Estadística (INE), www.ine.gub.u y. The map of Uruguay was obtained from USGS National Map Viewer (http://viewer.nationalmap.gov/viewer/), one of the resources included in your list. 

Additionally, in "SFigure1.tif" in our submission the authors created the images themselves.

4. Please include captions for your Supporting Information files at the end of your manuscript, and update any in-text citations to match accordingly.

We have included the captions for the Supporting Information files at the end of the manuscript an updated in-text citations. 

5- Wastewater surveillance was found to be a powerful and complementary approach to clinical testing in monitoring the SARS-CoV-2 pandemic. Authors report surveillance of variants of concern (VOCs) or No-VOCs agrees with clinical testing. It would be interesting to correlate these temporal changes with clinical data or the confirmed cases with statistical comparisons.

We have reviewed the literature and the publicly available epidemiological reports and research papers only report changes over time of VOCs (e.g gamma, delta). We have included those references in the paper. There are no references to amino acid changes in clinical cases. 

6 - Figure 2 amino acids and their frequencies are shown in circular graphs for each location. Is this a Pie chart? Presented in the form of a Pie should sum to 100%.

We have changed the representation of amino acids references in Figure 2.

7- The scale on the Y-axis for Figure 1C is not completely visible

Figure 1C has been modified to make the scale on the Y-axis completely visible.

8- It is somewhat surprising to see OK091006 used as a reference. While OK091006 indeed represents an example of the Delta variant of SARS-CoV-2 it is a Genbank record without extensive information on its provenance and quality control. The authors do not explain why this was chosen in addition to the more conventional Wuhan Hu-1 reference (which has been the subject of several revisions and has been deposited in NCBI's RefSeq collection of curated sequences)

We used the Wuhan Hu-1 and the Delta variant,OK091006, and similar results were obtained for both analyses. The OK091006, sequence was used because it has the same original variations as the Wuhan sequence (MN908947) and it also presents mutations associated with Delta (L452R, T478K), within at least the RBD region of the delta B.1.617.2 lineage. 

9- The authors claim that data is available in NCBI SRA but I could not find it using the accession numbers provided. If the data is embargoed pending publication the authors should state that.

The data is embargoed pending publication. We have added this statement to the manuscript.

---

## [Editor Report · Decision Letter 1]

3 Apr 2023

Wastewater surveillance of SARS-CoV-2 genomic populations on a country-wide scale through targeted sequencing

PONE-D-22-31923R1

Dear Dr. Trujillo,

We’re pleased to inform you that your manuscript has been judged scientifically suitable for publication and will be formally accepted for publication once it meets all outstanding technical requirements.

Kind regards,

Nagarajan Raju

Academic Editor

PLOS ONE
---

## [Editor Report · Acceptance letter]

12 Apr 2023

PONE-D-22-31923R1 

Wastewater surveillance of SARS-CoV-2 genomic populations on a country-wide scale through targeted sequencing 

Dear Dr. Trujillo:

I'm pleased to inform you that your manuscript has been deemed suitable for publication in PLOS ONE. Congratulations! Your manuscript is now with our production department. 

Kind regards, 

on behalf of

Dr. Nagarajan Raju 

Academic Editor

PLOS ONE